# Distribution and Characterization of Antimicrobial Resistant Pathogens in a Pig Farm, Slaughterhouse, Meat Processing Plant, and in Retail Stores

**DOI:** 10.3390/microorganisms10112252

**Published:** 2022-11-14

**Authors:** Dongryeoul Bae, Donah Mary Macoy, Waqas Ahmad, Son Peseth, Binn Kim, Jung-Whan Chon, Gyeong Ryul Ryu, Ga-Hee Ban, Sun Ae Kim, Hye Jeong Kang, Jin San Moon, Min Gab Kim

**Affiliations:** 1Research Institute of Pharmaceutical Science, College of Pharmacy, Gyeongsang National University, Jinju 52828, Korea; 2Garam Ltd., Eumseong-gun 27617, Chungcheongbuk-do, Korea; 3Companion Animal Health, Inje University, Gimhae 50834, Korea; 4Department of Food Science and Biotechnology, Ewha Womans University, Seoul 03760, Korea; 5Bacterial Disease Division, Animal and Plant Quarantine Agency, Gimcheon 39660, Korea

**Keywords:** foodborne pathogens, pork, environment, production, distribution

## Abstract

The emergence of antibiotic resistance in foodborne pathogens isolated from meat pro-ducts and their producing environment has been an increasing and leading threat to public health. The aim of the study was to identify pathogens and their antimicrobial resistance isolated from pig production to pork meat distribution phases. Through this study, food spoilage and foodborne or clinical pathogenic bacteria were isolated and identified from pork (belly and neck) meat product and its related environmental samples that include pig swabs, diets, feces, liquid manure, workers’ gloves, dust fan swabs, carcass swabs, floor swabs, and drain water in the affiliated farm, slaughterhouse, meat processing plant, and in retail stores. All carcasses at the slaughterhouse and meat products at the meat processing plant were tracked from pigs at a targeted farm. Nine different selective media agars were used to effectively isolate various pathogenic bacteria. A total of 283 presumptive pathogenic bacteria isolated from 126 samples were selected and identified using MALDI-ToF MS. Twenty-three important foodborne pathogens were identified, and some of them, Shiga-toxin-producing *E. coli* (STEC), *Listeria monocytogenes*, *Staphylococcus aureus*, and *Yersinia enterocolitica*, were further confirmed using PCR. The PFGE patterns of 12 STEC isolates were grouped by sample source or site. All the foodborne pathogens used in the study were not resistant to amoxicillin/clavulanate, ciprofloxacin, and gentamicin, whereas some of the STEC, *L. monocytogenes*, and *S. aureus* isolates were resistant to various antibiotics, including ampicillin, erythromycin, tetracycline, and vancomycin. The most common antimicrobial resistance pattern in the pathogenic STEC isolates was AMP-KAN-STR-SXT-TET. Consequently, this study provides valuable information for the distribution of antimicrobial-resistant pathogens along the pork meat production chain and can assist farmers and stakeholders to develop a systematic strategy for reducing the current emergence and spread of antimicrobial resistance in the different phases of pig production and distribution.

## 1. Introduction

The increased demand for animal protein according to the steady growth of the world population has resulted in the expansion of the pig population and large-sized pig farms. The increased use of antimicrobials at the farms, particularly in the phase of weaning, may have accompanied increased antimicrobial-resistant bacteria [1,2,3]. As a result, the increased antimicrobial resistance (AMR) can be a burden for human and animal health, since AMR can be transferred between humans, animals, and their environment through horizontal gene transfer by transformation or conjugation [4,5]. In addition, the prevalence of the pathogens with multidrug resistance (MDR) is raising serious concerns for human and animal health and welfare through the limitation of antimicrobials available to treat diseases. These issues can be also important for food safety and security, as well as global trade [6]. The production of pork meat is globally expanding due to economical and nutritional values. Therefore, the inhibition of increased antimicrobial-resistant bacteria in pork meat and its byproducts is one of the main concerns for farmers, food policymakers, stakeholders, and consumers, and is highlighted as there is an increased demand for the consumption of the safe meat products. The development of AMR is a typical evolutionary process in microbes, and the increasing use of antibiotics has led to the emergence of AMR, which limits the treatment of various bacterial infections [7]. Currently, extensive efforts have been conducted in South Korea to minimize AMR. These include the Korean National Antimicrobial Resistance Safety Control Program headed by the Korean national action plan on AMR, ruled by the Korean Ministry of Health and Welfare from 2016 to 2020 and the Korean Ministry of Food and Drug Safety from 2003 to 2013 [8].

The use of antimicrobials in the livestock sector for over 60 years has resulted in the dissemination and coselection of antimicrobial-resistant bacteria in food-producing animals [9,10]. Consequently, the transmission of antimicrobial-resistant traits or bacteria between humans and livestock, and the associated environment, has been frequently present [11]. Currently, the therapeutic use of antimicrobials is allowed to treat diseased animals, whereas the use of antimicrobials for the growth promotion and prophylactic purposes is strictly prohibited in most developed countries [12]. However, antimicrobials are still used for this purpose in many developing countries due to the lack of legislative systems and veterinary infrastructure [2,13]. Recently, foodborne pathogens with high MDR in the phases from pork production to distribution have emerged [14,15,16]. Lunha and colleagues showed that 23% of the *E. coli* isolates from healthy pigs’ feces in the medium-scale commercial swine farms in northeastern Thailand were shown to be MDR, with the most common pattern being CHL-TET-SXT. Another study presented that various foodborne pathogens, including *Salmonella* spp., *Clostridium perfringens*, and *Staphylococcus aureus*, were isolated in edible pig offal samples from 11 pig slaughterhouses in S. Korea [17]. Recently, antimicrobial-resistant bacteria, including methicillin-resistant *S. aureus* (MRSA), *Salmonella* spp., and extended-spectrum β-lactamase-producing (ESBL) *Enterobacteriaceae*, were isolated from fresh and processed retail pork meat products [18].

In the United States and S. Korea, *Salmonella* as a potential microbiological hazard in carcasses and pork meat products along with other pathogens are primarily monitored to confirm the existence of the pathogenic isolates and assess the hygiene status of pork meats. According to the CDC, the most frequent pathogens associated with outbreaks from 1998 to 2015 through the consumption of pork products were *Salmonella*, *S. aureus*, *C. perfringens*, *Bacillus cereus*, *Y. enterocolitica*, and *Shigella* [19]. Additionally, those pathogens with MDR have been threatening both human and animal health [20]. Therefore, making an effort to reduce AMR should be performed globally and systematically. With continued efforts, monitoring the prevalence of pathogens in pork products and their producing environment is also necessary to make a preventive plan for reducing AMR bacteria. Until now, most studies have fragmentarily monitored the emergence of the foodborne pathogens from each of the farm, slaughterhouse, meat processing plant, or retail store. Accordingly, the purpose of the current study was to examine whether various pathogens with AMR in the pork meats exist and transmit from their producing phase (at the farm, slaughterhouse, and meat processing plant levels) to their distribution phase (at retail meat and grocery stores).

## 2. Materials and Methods

### 2.1. Sampling and Treatment

Pork and samples from the production environment were collected from a midsized swine farm (holding 5000 pigs), and the subsequent slaughterhouse, meat processing plant, and retail stores located in Gyeongki-do, South Korea. Pig samples transferred from different locations, such as from the farm to the slaughterhouse, then to the meat processing plant, and finally the processed meat transported to the retail stores, were strictly tracked using the Korean animal product tracking system (animal products traceability, https://mtrace.go.kr/, accessed on 21 May 2021). A total of 126 samples (*n* = 126) were collected from at least three pigs, carcasses, meats, and their related environment (Appendix A). Three finishing pig skins’ swab, diet, feces, liquid manure, floor swab, workers’ gloves, and dust fan swab samples were collected from three pig barns. Each of the three carcass swabs, floor swabs, gloves, carcass halves knife swabs, washing water, and drain water samples were collected from the slaughterhouse. In addition, three processed meat swabs, floor swabs, water, table swabs, gloves, and drain water samples were collected from the meat processing plant, and three pork belly and neck meat products were obtained from three meat retail stores. All samples were sequentially collected from the farm to markets for 4 consecutive days in May 2021 to track the transmission of pathogens to meat samples from their production environments. In detail, the carcass, processed meat, and market meat samples collected from each phase were obtained from the pigs raised from the swine farm. Carcasses, processed meat, meat product, and other swab samples were collected using Whirl-Pak^®^ Speci-Sponge^®^ Environmental Surface Sampling Bag (Nasco, Fort Atkinson, WI, USA) according to the manufacturers’ directions. The areas swabbed at the back of the carcasses and the meats at the meat processing plant were approximately 400 cm^2^. Water and drain water samples were collected using 2 L sterilized sampling bottles (Mediland, Seoul, Korea) and 50 mL conical centrifuge tubes (SPL Lifesciences Co., Ltd., Pocheon, Korea), respectively. A new sterilized sponge, sampling bag, and glove were used for collecting each swab sample. All samples collected for the current study were placed in the icebox and transported to the laboratory within 4 h. An amount of 100 g of meat products from the retail stores, 25 g of feces, and other environmental swab samples were rinsed using 225 mL buffered peptone water (Oxoid, Hampshire, United Kingdom), and rinsates and water and drain water samples were placed in 3M^TM^ filtered sample bags (3M, St. Paul, MN, USA) to remove the insoluble residues. The filtered samples were then centrifugated at maximum speed (7193× *g*), the pellets were dispersed with 3 mL of phosphate buffered saline (Biosesang, Gyeonggi-do, Korea), and the concentrated samples were stored at −4 °C until use.

### 2.2. Pathogen’s Isolation and Identification

One hundred microliters of watery samples or rinsates stored at refrigerator were directly spread on various selective (Tryptose Sulfite Cycloserine (TSC), Oxford, MacConkey, Xylose Lysine Deoxycholate (XLD), Tellurite Cefixime-Sorbitol MacConkey (TC-SMAC), 5-Bromo-4-Chloro-3-Indolyl-β-D-Glucuronide (BCIG), Mannitol Egg Yolk Polymyxin (MYP), Baird–Parker, and modified Charcoal Cefoperazone Deoxycholate (mCCDA)) and Brain Heart Infusion (BHI) growth agar plates and incubated aerobically or anaerobically at 30 °C to 42 °C for 24 h to 48 h to effectively isolate major foodborne pathogens, including *C. perfringens*, *Listeria monocytogenes*, nontyphoidal *Salmonella* spp., *Yersinia enterocolitica*, pathogenic *E. coli*, *Staphylococcus aureus*, and *Campylobacter jejuni*/*coli*. If needed, the samples were serially diluted. About one to ten single colonies from each of the selective agar plate were picked up, the single colony was inoculated in BHI broth (Becton Dickinson, Sparks, MD, USA), and the cultures mixed with 40% of glycerol were stored at −80 °C until use. The freshly grown pathogens were identified using the Matrix-Assisted Laser Desorption/Ionization-Time of Flight Mass Spectrometry (MALDI-TOF MS; Bruker Daltonics, Bremen, Germany) and the MALDI-TOF MS Biotyper 3.1 software. A total of 283 of 1, 400 colonies were identified as presumptive pathogens or spoilage bacteria, and 23 important foodborne pathogens (1 *L. monocytogenes*, 9 *Y. enterocolitica*, 12 Shiga-toxin-producing *E. coli* (STEC), 1 *S. aureus*) were then confirmed using PCR with the primer sets (Table 1).

### 2.3. Antimicrobial Susceptibility Assay

Antimicrobial susceptibility tests (ASTs) on the major foodborne pathogens using a disc diffusion assay were evaluated according to the Clinical and Laboratory Standards Institute guidelines [26]. Eleven antibiotics, including ampicillin (AMP), amoxicillin/clavulanate (AMC), chloramphenicol (CHL), ciprofloxacin (CIP), erythromycin (ERY), gentamicin (GEN), kanamycin (KAN), streptomycin (STR), tetracycline (TET), rifampicin (RIF), and trimethoprim-sulfamethoxazole (SXT), were used for Gram-negative bacteria. Vancomycin (VAN) was additionally used for Gram-positive bacteria. The purely isolated pathogens’ colonies were cultured in Mueller Hinton (MH) media (Becton Dickinson). The cultures were measured 0.15 at OD600 using the SpectraMax^®^ microplate spectrophotometer (Molecular Devices, San Jose, CA, USA). MH agar plates and antibiotic diffusion assay discs (Becton Dickinson) were used. Antibiotic resistant, intermediate, and susceptible assessment were determined according to the manufacturers’ instructions. *S. aureus* (ATCC 25923) and *E. coli* (ATCC 29212) were used as control strains.

### 2.4. Pulsed-Field Gel Electrophoresis (PFGE) Analysis

PFGE was performed for genetic characterization of the 12 STEC isolates using the standard operating procedure for U.S. Centers for Disease Control and Prevention PulseNet PFGE of *E. coli* (http://www.cdc.gov/pulsenet/PDF/listeria-pfge-protocol-508c.pdf, accessed on 22 September 2022), slightly modified. In brief, bacterial isolates were grown in LB broth at 37 °C for 16 h. Two-hundred microliters of cultures were adjusted to an OD610 of 0.43 for making plugs. The adjusted cultures were gently mixed with 200 μL of 1.0% agarose (SeaKem Gold Agarose, Cambrex, ME, USA) and 10 μL of proteinase K stock solution (20 mg/mL). Bacterial cells in the plugs were lysed, washed, and stored at 4 °C until use. The plugs were cut to 2 mm. The cut plug slices containing bacterial DNA were placed into and digested in 1.5 mL microcentrifuge tubes containing *Xba*I (New England Biolabs, Ipswich, MA, USA) restriction buffer (50 U per plug slice) for 4 h at 37 °C. The restricted PFGE plugs were finally washed, loaded into 1% SeaKem Gold agarose gels, and electrophoresed using CHEF Mapper (Bio-Rad Laboratories, Hercules, CA, USA) at 6 V/cm for 19 h. The initial and final switch times were 2.16 s and 54.17 s, respectively. Images were obtained by Molecular Analyst software version 1.1 (Bio-Rad Laboratories). The PFGE patterns for DNA finger printing were analyzed by Bionumerics software v.8.1 (Applied Maths, Inc., Austin, TX, USA). The similarity of PFGE banding patterns for STEC isolates was determined using unweighted pair group method with arithmetic mean (UPGMA).

## 3. Results

### 3.1. Distribution of the Pathogens in the Pork Meats and Their Producing Environmental Samples

Pathogenic bacteria isolated from the pork production chain and corresponding environment samples were identified and confirmed using MALDI-TOF MS and PCR, respectively. From 126 samples, 1400 colonies were collected, and 283 of them were identified as belonging to the bacteria focused on in this study (Figure 1). Four major foodborne pathogens, STEC, *S. aureus*, *L. monocytogenes*, and *Y. enterocolitica*, were found in the study. Particularly, nine STEC were isolated from belly meat at a high rate (Figure 1). Appendix A shows that 12 STEC were distributed in feces at the pig farm, on carcasses and gloves at the slaughterhouse, and in belly meats at the retail stores. Other *E. coli* isolates were found in feces and liquid manure at the farm, the working table at the meat processing plant, and the neck meat at the retail stores. *L. monocytogenes* and *S. aureus* were recovered from the pork meat and the working table at the meat processing plant, respectively (Appendix A). Ten *Y. enterocolitica* were mostly found in the slaughterhouse (carcasses, workers’ gloves, and cutting knife). Besides the major foodborne pathogens, the clinically important pathogens associated with the WHO global priority pathogens list of antibiotic-resistant bacteria included *Acinetobacter baumannii*, *Enterobacter cloacae*, *Morganella morganii*, *Proteus mirabilis*, *Providencia rustigianii*, and *Serratia liquefaciens*, found in various sampling sites from swine production to the distribution phase (Appendix A). Of the bacteria identified, isolates of *Pseudomonas lundensis*, *S. liquefaciens*, and *Staphylococcus sciuri* isolates were most abundant in the meat and environments. *S. saprophyticus* and *S. sciuri* isolates were found in all the phases of production, slaughtering, meat processing, and distribution, whereas *A. baumannii*, *Ewingella americana*, *Kocuria rhizophila*, *P. mirabilis*, *Providencia rustigianii*, *Pseudomonas antarctica*, *Pseudomonas synxatha*, and *Pseudomonas tolaasii* isolates were found only in the pork belly or neck meat products at the distribution phase. There were no pathogenic bacteria in the water used in the slaughterhouse and meat processing plant.

### 3.2. Antimicrobial Resistance in the Important Foodborne Pathogens

Foodborne pathogens such as *Y. enterocolitica*, *STEC*, *S. aureus*, and *L. monocytogenes* were tested for antimicrobial susceptibility. Twelve and eleven different antibiotic disks used in the study were used for Gram-positive and Gram-negative bacteria, respectively (Table 2). VAN was not used for Gram-negative bacteria due to inefficiency. Data showed that most of the *Y. enterocolitica* isolates were susceptible to the antimicrobials used in this study (Table 2). Only one pathogen, *Y. enterocolitica* (GNU-YE6), was resistant to ampicillin and intermediate to ERY and RIF. GNU-YE5 and GNU-YE7 isolates were intermediate to AMC, AMP, ERY, and RIF resistance. There were no multidrug-resistant *Y. enterocolitica* isolates. On the other hand, most of the STEC isolates were resistant to multiple drugs, including AMP, CHL, ERY, KAN, STR, SXT, and TET. In addition, the most common antimicrobial resistance pattern in the pathogenic *E. coli* isolates was AMP-KAN-STR-SXT-TET. *L. monocytogenes* and *S. aureus* were also shown to be MDR to AMP, ERY, and VAN. All the important foodborne pathogens were susceptible to CIP and GEN.

### 3.3. Genetic Diversity of STEC Isolates

Genomic DNA of 12 STEC isolate strains were digested with *Xba*I to determine genetic relatedness of the isolates for tracking the pathogens from farm to distribution. The isolates were classified into four pulsotypes based on PFGE banding patterns with more than 99% similarity (Figure 2). The dendrogram of PFGE profiles by *Xba*I-digestion shows that all STEC isolates grouped in pulsotype I were isolated from belly meats at the retail stores. On the other hand, the other isolates (GNU-EC1, GNU-EC2, and GNU-EC3) showed exceedingly different PFGE band patterns by sample source or site.

## 4. Discussion

As far as we know, this study is conducted for the first time to track pathogens from pigs and the pork-meat-producing environment to pork meat products via the affiliated slaughterhouse and meat processing plant where the targeted pigs were treated and traced. Thus, this study provides valuable information for the transmission of pathogens between pork meat products and their producing environment. Nine selective agar plates were used to efficiently isolate various bacteria in the samples. Although many colonies with the right form and color to presumably expected pathogens on TSC, XLD, MYP, and mCCDA agar plates were picked up and identified, highly important foodborne pathogens, including *Clostridium perfringens, Salmonella* spp., *Bacillus cereus,* and *Campylobacter jejuni*/*coli*, were not found in this study. Nevertheless, we found that various foodborne pathogens are present in the pork meats and their producing-related environment. STEC isolates of the pathogens were sequentially found in the farm, slaughterhouse, and retail meat stores, unlike the other pathogens (Appendix A). Through subtyping for the STEC colonies, we found that the STEC strains (pulsotype 2, 3, and 4) isolated in the samples from the farm and slaughterhouse environment and carcasses seemed not to be transmitted into the pork meats due to different pulsotypes (Figure 2). STEC strains found in the pork meats were likely to be contaminated from the workers or environments at the retail stores.

The STEC isolates from the pork meat products in retail stores were resistant to various antimicrobials (AMP, KAN, STR, SXT, and TET) than those that at farm and slaughterhouse. A previous report presented that more than 50% of *E. coli* isolates from fecal and carcass samples were resistant to AMP, CHL, STR, SUL, and TET. In the study, STEC GNU-EC 1 and GNU-EC 2 strains were shown to be resistant to AMP and ERY and AMP and CHL, respectively. In addition, the strains were intermediate to TET (Table 2). The resistance of these antibiotics would be closely associated with the annual antibiotic consumption in the pig industry in S. Korea because antibiotics in the penicillin, tetracycline, and phenicol classes are widely used [3].

With the current results, *Y. enterocolitica* has been dominantly isolated from food-chain animals and meats, and the bacteria have exhibited to be resistant to various antibiotic classes, including penicillin, cephalosporin, macrolide, aminoglycoside, fluoroquinolone, and tetracycline [27,28,29]. The Korean Ministry of Food and Drug Safety recently reported that *Y. enterocolitica* was found in 15 out of 289 kimchi products imported from China (www.aninews.in/news/world/asia/bacteria-causing-plague-detected-in-korean-delicacy-kimchi-exported-from-china20210520232537/, accessed on 29 July 2022). It was magnified as an important issue in the food safety sector in S. Korea, since the pathogen, causing gastroenteritis, was reported to cause 35 deaths annually in the US [30]. The pathogens were isolated from the carcasses, the workers’ gloves, and the cutting knife at slaughterhouse and the pork belly meat product (Appendix A). In addition, the isolates were susceptible to almost all antibiotics used in the study, except one isolate resistant to AMP (Table 2). *P. mirabilis* isolates have shown to have different antimicrobial resistance patterns to CHL, ERY, and TET (data not shown). The pathogen has been known to cause urinary tract infection and food poisoning outbreaks [31,32,33]. *P. mirabilis* is a normal flora in the digestive tracts of humans and animals and is widely found in the natural environment, including soil and water [32]. However, many foodborne poisoning cases associated with the pathogen were recently reported worldwide via nosocomial infection and contaminated food ingestion [34]. Although *P. mirabilis* isolates were not found in the farm, slaughterhouse, and meat processing plant, the pathogen was found in the pork meat products. It is suggested that the pathogen may be cross-contaminated at the stores. We failed to collect environmental samples in the stores due to exceed repulsion of the owners for collecting the samples. This can be a drawback of the current study. A recent study showed that 5.63% pork samples were contaminated with *P. mirabilis* producing β-lactamase [35]. However, all *P. mirabilis* isolated in the study were susceptible to β-lactams. Van was not used for the antibiotic test for Gram-negative pathogens, since VAN has large molecular size and inability to penetrate the outer membrane of the bacteria.

With respect to prevention of the spread of AMR, proper and prudent use of antimicrobials for the pig production is imperative to minimize the emergence and development of bacteria resistant to clinically important antibiotics. In addition to the endeavor to restrict the antibiotic use, monitoring the AMR patterns of clinically important pathogens is also an important step to ensure the safety of meat products and public health. Through the current study, we have tracked clinically important foodborne pathogens from farm to table. Some of the pathogenic or clinically important bacteria, including STEC, *Proteus vulgaris*, *Serratia liquefaciens*, and *Y. enterocolitica*, were found in the pork meat products as well as the farm, slaughterhouse, or meat processing plant. These bacteria contaminated with the pork meats can be transmitted from farm to the meat processing environment. However, other pathogens, such as *Acinetobacter baumannii*, *P. mirabilis*, and *Providencia rustigianii*, were only found in the pork meat products, suggesting that the pathogens would be transmitted via cross contamination at the retail stores. As previously mentioned, the study was conducted for the first time wherein pathogens were tracked from targeted pigs to processed meat products with the pigs’ carcasses, as well as their environment in which the pathogens possibly reside. Consequently, the present study may provide valuable information for presumable sites that clinically important pathogens can be highly transmitted to the pork meat products during the slaughtering and processing phases. Furthermore, the results may help farmers, stakeholders, and government to develop a systematic strategy for reducing the emergence and spread of antimicrobial resistance in pork meat and its producing environment to ensure meat safety and ultimately promote public health.

## Figures and Tables

**Figure 1 microorganisms-10-02252-f001:**
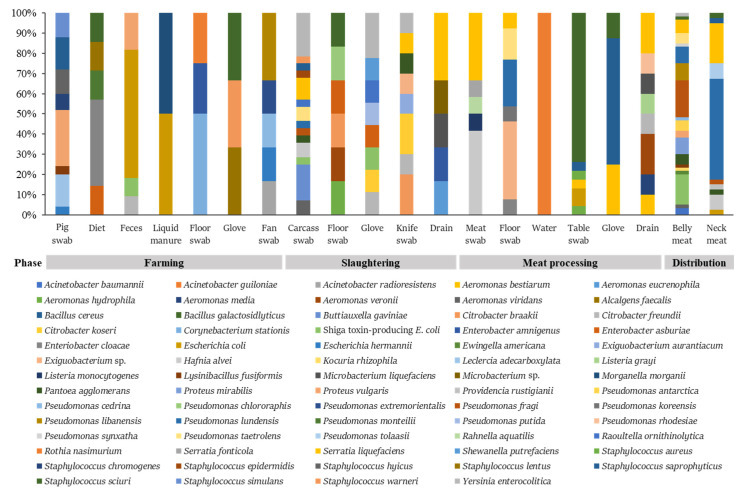
The ratio of pathogens isolated from the pork meat and its producing environment. Bacteria isolated from various samples were identified using the MALDI-TOF MS and the MALDI-TOF MS Biotyper 3.1 software. Important foodborne pathogens, including STEC, *Listeria monocytogenes*, *Staphylococcus aureus*, and *Y. enterocolitica* were found in the pork meat products and their producing environment from the farm, slaughterhouse, or meat processing plant. Additionally, clinically important pathogens (e.g., *Acinetobacter baumannii*, *E. coli*, *Enterobacter* spp., *Serratia* spp., *Proteus mirabilis*, *Providencia rustigianii*, and *Morganella morganii*) highly related to antimicrobial resistance were found in in the pig farms, slaughterhouse, meat processing plant, and retail stores.

**Figure 2 microorganisms-10-02252-f002:**
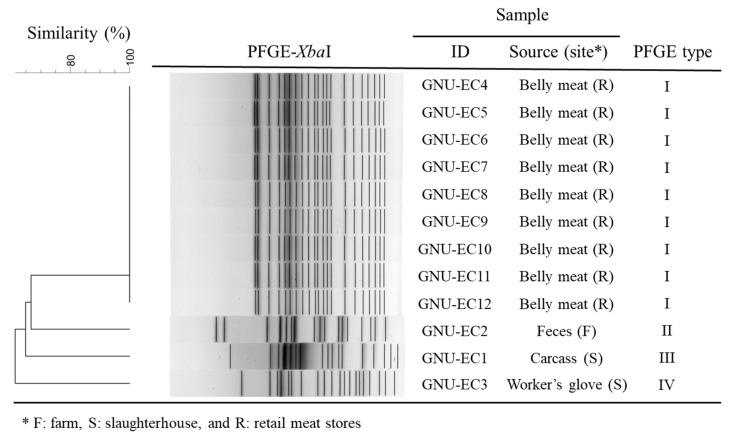
Analysis of PFGE banding patterns of STEC strains isolated from feces at swine farm, carcass, and the workers’ gloves at meat processing plant, and belly meats at retail stores. Dendrogram of the PFGE patterns from 12 STEC meat and environmental isolates using *Xba*I-digestion is shown by 4 pulsotypes according to the similarity of PFGE band profiles of more than 99% within a group using UPGMA clustering method.

**Table 1 microorganisms-10-02252-t001:** Selective agars, cultivation conditions, and primer sequences used in the study.

Selective Agar	Target Bacteria	Characteristics/Cultivation Conditions	Primer Sequences	Gene/Product Size	References
Oxford	*Listeria monocytogenes*	Gray colony with a black halo, 37 °C for 24 h	F: TGCAAGTCCTAAGACGCCA R: CACTGCATCTCCGTGGTATACTAA	*hlyA*/113 bp	[21]
MacConkey	*Yersinia enterocolitica*	Pale pink or colorless colony, 30 °C for 36 h	F: TTTGGAAGCGGGTTGAATTG R: GCTCACGGAAAGGTTAAGTCATCT	*ail*/101 bp	[22]
TC-SMAC	STEC	Clear or colorless colony, 37 °C for 24 h	F: ATAAATCGCCATTCGTTGACTAC R: AGAACGCCCACTGAGATCATC F: GGCACTGTCTGAAACTGCTCC R: TCGCCAGTTATCTGACATTCTG	*stx1*/180 bp *stx2*/255 bp	[23]
BCIG	*E. coli*	Bluish green colony, 37 °C for 24 h	F: CCGATACGCTGCCAATCAGT R: CTGGTATCAGCGCGAAGTCT	*uspA*/884bp	[24]
Baird-Parker	*Staphylococcus aureus*	Gray to less black with opaque zone, 35 °C for 48 h	F: AAGTGCCGATCAATTTATGGCTA R: CCTGAACAGTTACATTTTTCTTATTCGT	*entA*/90 bp	[25]

**Table 2 microorganisms-10-02252-t002:** Antimicrobial ^1^ susceptibility test of major pathogens using disc diffusion assay.

ID	Species	AMC	AMP	CHL	CIP	GEN	ERY	KAN	RIF	STR	SXT	TET	VAN	Source (Site ^2^)
GNU-YE 1	*Y. enterocolitica*	−	−	−	−	−	−	−	−	−	−	−	NA	Carcass (S)
GNU-YE 2	*Y. enterocolitica*	−	−	−	−	−	−	−	−	−	−	−	NA	Carcass (S)
GNU-YE 3	*Y. enterocolitica*	−	−	−	−	−	−	−	−	−	−	−	NA	Carcass (S)
GNU-YE 4	*Y. enterocolitica*	−	−	−	−	−	−	−	−	−	−	−	NA	Carcass (S)
GNU-YE 5	*Y. enterocolitica*	+	+	−	−	−	+	−	+	−	−	−	NA	Glove (S)
GNU-YE 6	*Y. enterocolitica*	+	++	−	−	−	+	−	+	−	−	−	NA	Knife (S)
GNU-YE 7	*Y. enterocolitica*	+	+	−	−	−	+	−	+	−	−	−	NA	Glove (S)
GNU-YE 8	*Y. enterocolitica*	−	+	−	−	−	−	−	−	−	−	−	NA	Belly meat (R)
GNU-YE 9	*Y. enterocolitica*	−	−	−	−	−	−	−	−	−	−	−	NA	Carcass (S)
GNU-EC 1	STEC ^3^	−	++	−	−	−	++	−	−	−	−	+	NA	Carcass (S)
GNU-EC 2	STEC	−	++	++	−	−	+	−	−	+	−	+	NA	Feces (F)
GNU-EC 3	STEC	−	++	++	−	−	+	−	+	+	++	+	NA	Glove (S)
GNU-EC 4	STEC	−	++	+	−	−	+	++	+	++	++	++	NA	Belly meat (R)
GNU-EC 5	STEC	−	++	+	−	−	++	+	+	++	+	++	NA	Belly meat (R)
GNU-EC 6	STEC	−	+	+	−	−	+	++	+	++	++	+	NA	Belly meat (R)
GNU-EC 7	STEC	−	++	+	−	−	+	++	+	++	+	++	NA	Belly meat (R)
GNU-EC 8	STEC	−	++	+	−	−	+	++	+	++	+	++	NA	Belly meat (R)
GNU-EC 9	STEC	−	++	+	−	−	+	++	+	++	++	++	NA	Belly meat (R)
GNU-EC 10	STEC	−	++	+	−	−	+	++	+	++	++	++	NA	Belly meat (R)
GNU-EC 11	STEC	−	++	+	−	−	+	++	+	++	++	++	NA	Belly meat (R)
GNU-EC 12	STEC	−	++	+	−	−	++	++	+	++	++	++	NA	Belly meat (R)
GNU-SA 1	*S. aureus*	−	++	−	−	−	++	−	+	−	−	+	++	Table (P)
GNU-LM 1	*L. monocytogenes*	−	++	−	−	−	++	−	−	−	−	+	++	Meat (P)

++, Resistant; +, intermediate; −, susceptible; ^1^ AMC: amoxicillin and clavulanic acid, AMP: ampicillin, CHL: chloramphenicol, CIP: ciprofloxacin, ERY: erythromycin, GEN: gentamicin, KAN: kanamycin, RIF: rifampicin, STR: streptomycin, SXT: trimethoprim and sulfamethoxazole, TET: tetracycline, VAN: vancomycin; ^2^ F: farm, S: slaughterhouse, P: meat processing plant, and R: retail meat stores; ^3^ Shiga-toxin-producing *E. coli.*

## Data Availability

Not applicable.

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
