# Peer review of "Distribution and Characterization of Antimicrobial Resistant Pathogens in a Pig Farm, Slaughterhouse, Meat Processing Plant, and in Retail Stores"

_microorganisms, 2022, doi:10.3390/microorganisms10112252_

Round 1
Reviewer 1 Report
In the study, pathogens were isolated from samples taken at subsequent stages of the pork production chain from farm to retail. The isolates were identified and tested for their antimicrobial resistance. Despite the animals and followin pork products, also a variety of samples from the production environment were investigated. Due to the S. Korean tracking system it was possible to follow the same animal batches and products thereof from farm to retail. This is an interesting approach and valuable insight to the spread of bacteria along the pork production chain can be expected.
The manuscript in gereral is well structured, but needs proofreading by a native speaker. Escpecially the discussion needs to be rephrased.
The description of the method used for resistance testing is inconclusive. Therefor the comparability of the results with other studies is restricted, which should be stated in the discussion. In detail: For the method, citation 21 refers to CLSI M100, which does not provide the general methods for testing, but only the criteria for evaluation of the results. In line it is 154 stated that the concentration used for the disc diffusion was 0.5 at OD600. This is way above the CLSI specifications for non-fastidious bacteria (McFarland 0.5, which corresponds to 0.08-0.1 at OD600). For the evaluation of the results it is referred to the manufacturer’s instructions, but here it is not clear whether this is valid for the method used. Do they demand the concentration of 0.5 at OD600? Please correct the description of the method.
The discussion on the transmission of distinct bacterial clones along the pork production chain would become more significant with addition of clonal analyses of the E. coli O157:H7 isolates. I suggest adding this additional analyses as it would be a valuable add-on to the study.
Some suggestions for improvement of specific text passages:
Line 24: Please change to ‘A total of 283 isolates of pathogenic bacteria…
Line 32: Please change to ‘along the pork meat production chain’
Line 66-67: Please change to e.g. ‘However, antimicrobials are still used for this purpose in many developing countries…’
Line 73: Please insert ‘including’ before ‘Salmonella’
Line 96: Please rephrase. E.g. ‘Pork and samples from the production environment were collected from a mid-size swine farm (…), and the subsequent slaughterhouse, meat processing plant and retail stores…’
Line 135: Please delete ‘spp.’ after E. coli, add ‘E. coli O157:H7’ as a separate entry
Line 136: Please replace ‘they’ by ‘the samples’
Line 161: Please rephrase to ‘Pathogenic bacteria isolated from pork production chain and corresponding environment samples….’
Line 162-164: Please rephrase. E.g. ‘From 126 samples 1,400 colonies were isolated, 283 of them identified as belonging to the bacteria focused in this study (Figure 1 and Table 1).’
Lines 177-178: Please rephrase. E.g. ‘Of the bacteria identified, isolates of … were most abundant in meat and associated environments.’
Line 196: Please add ‘resistant’ after ‘MAR’
Line 197: Please replace multi-drugs by ‘multiple drugs (MDR)’
It was very difficult for me to follow the discussion and therefore I do not see myself beeing able making specific modifications. I suggest rephrasing of the entire section, if possible with the assistance of a native speaker
Line 244-246: Add the information on the lack of comparability of the resistance results of the present study with other studies due to deviation from the standard method.
Line 247: What is ‘CHR’ standing for?
Author Response
11-03-2022
Academic Editor
Microorganisms
Dear Academic Editor:
Enclosed is our latest version of microorganisms-1873710, that is, the revision of our paper.
We would like to take this opportunity to express our thanks to the reviewers for the positive feedback and helpful comments for correction or modification.
We believe that the given suggestions have resulted in an improved revised manuscript underlined, which you will find uploaded alongside with this document. The manuscript has been revised to address the comments from reviewers, which are appended alongside our responses to this letter.
We very much hope that the revised manuscript will be accepted for publication in Microorganisms.
Sincerely yours,
Dongryeoul Bae
Mailing address: Research Institute of Pharmaceutical Science, College of Pharmacy, Gyeongsang National University, Jinju 52828, S. Korea.
Phone: +82-55-772-2416; E-mail: [email protected]
Min Gab Kim
Mailing address: Research Institute of Pharmaceutical Science, College of Pharmacy, Gyeongsang National University, Jinju 52828, S. Korea.
Phone: +82-55-772-2427; E-mail: [email protected]
Manuscript No: Microorganisms-1873710
We thank the reviewers for their comments and suggestions on our manuscript. The revised manuscript has been edited taking into the account of all suggestions by the reviewers. In addition, we changed table 1 to supplementary data table 1 and modified table 2 and 3. We focused on foodborne pathogens and their isolation and identification methods. Figure 2 for analysis of PFGE banding patterns of STEC isolates was added. E. coli O157:H7 was replaced to Shiga toxin-producing E. coli (STEC). We believe that the revised manuscript is suitable for publication in Microorganisms. Our responses to the reviewer’s questions are listed below:
Reviewer #1
Comments to the Author
In the study, pathogens were isolated from samples taken at subsequent stages of the pork production chain from farm to retail. The isolates were identified and tested for their antimicrobial resistance. Despite the animals and following pork products, also a variety of samples from the production environment were investigated. Due to the S. Korean tracking system it was possible to follow the same animal batches and products thereof from farm to retail. This is an interesting approach and valuable insight to the spread of bacteria along the pork production chain can be expected.
The manuscript in gereral is well structured, but needs proofreading by a native speaker. Escpecially the discussion needs to be rephrased.
The description of the method used for resistance testing is inconclusive. Therefor the comparability of the results with other studies is restricted, which should be stated in the discussion. In detail: For the method, citation 21 refers to CLSI M100, which does not provide the general methods for testing, but only the criteria for evaluation of the results. In line it is 154 stated that the concentration used for the disc diffusion was 0.5 at OD600. This is way above the CLSI specifications for non-fastidious bacteria (McFarland 0.5, which corresponds to 0.08-0.1 at OD600). For the evaluation of the results it is referred to the manufacturer’s instructions, but here it is not clear whether this is valid for the method used. Do they demand the concentration of 0.5 at OD600? Please correct the description of the method.
The discussion on the transmission of distinct bacterial clones along the pork production chain would become more significant with addition of clonal analyses of the E. coli O157:H7 isolates. I suggest adding this additional analyses as it would be a valuable add-on to the study.
Response: We appreciate your careful review and suggestions. We have addressed all the specific comments and changed many parts in the revised version as you suggested. DNA fingerprinting of pathogenic E. coli isolates were performed to track the isolates from swine production to distribution phase using PFGE analysis. An English native speaker carefully revised this manuscript with proper English usage. In addition, we corrected ‘0.5 at OD600’ to ‘0.15 at OD600’. We deeply appreciate your exact indication.
Specific comments:
- L24 Please change to ‘A total of 283 isolates of pathogenic bacteria…’
We have changed as suggested. Ln 24-25.
- L32 Please change to ‘along the pork meat production chain’
We have changed as suggested. Ln 36.
- L66-67 Please change to e.g. ‘However, antimicrobials are still used for this purpose in many developing countries…’
We have changed as suggested. Ln 71-72.
- L73 Please insert ‘including’ before ‘Salmonella’
We have inserted as suggested. Ln 78.
- L96 Please rephrase. E.g. ‘Pork and samples from the production environment were collected from a mid-size swine farm (…), and the subsequent slaughterhouse, meat processing plant and retail stores…’
We have rephrased as suggested. Ln 102-104.
- L135 Please delete ‘spp.’ after E. coli, add ‘E. coli O157:H7’ as a separate entry
We have changed E. coli to pathogenic E. coli as suggested. Ln 146.
- L136 Please replace ‘they’ by ‘the samples’
We have replaced as suggested. Ln 147.
- L161 Please rephrase to ‘Pathogenic bacteria isolated from pork production chain and corresponding environment samples….’
We have rephrased as suggested. Ln 193-194.
- L162-164 Please rephrase. E.g. ‘From 126 samples 1,400 colonies were isolated, 283 of them identified as belonging to the bacteria focused in this study (Figure 1 and Table 1).’
We have rephrased as suggested. Ln 195-198.
- L177-178 Please rephrase. E.g. ‘Of the bacteria identified, isolates of … were most abundant in meat and associated environments.’
We have rephrased as suggested. Ln 214-216.
- L196 Please add ‘resistant’ after ‘MAR’
We have modified to ‘multidrug-resistant-‘. Ln 233-234.
- L197 Please replace multi-drugs by ‘multiple drugs (MDR)’
We have replaced as suggested. Ln 236.
- L244-246 Add the information on the lack of comparability of the resistance results of the present study with other studies due to deviation from the standard method.
We have rephrased as suggested. Ln 302-311.
- L247 What is ‘CHR’ standing for?
We have corrected ‘CHR’ to ‘CHL’. Ln 306.

Reviewer 2 Report
Target genes for identifying pathogen type (table 2) need to be named. it is difficult to know what gene is being used based on the primer sequence alone.
line 24 and other locations in the manuscript - the authors refer to 283 pathogens were detected, though the wording here should be changed to clarify that 283 of the isolates were presumptive pathogens, not that 283 different pathogens were found.
line 173-174 - please provide more details on why microbes such as Pantoea agglomerans are considered clinically important pathogens
one of the stated goals of this work was to track pathogens from farm to distribution. the methods used do not allow for tracking of specific subtypes, which would be necessary to achieve this goal. On lines 237 to 240, the authors state that the pathogens with different ABX profiles are likely to be different subtypes, but is not clear why that is. if mobile elements are responsible for encoding the resistance, why would the assumption be that the subtypes would be different? the same MLST subtype could have or not have the same ABX profile.
Author Response
11-03-2022
Academic Editor
Microorganisms
Dear Academic Editor:
Enclosed is our latest version of microorganisms-1873710, that is, the revision of our paper.
We would like to take this opportunity to express our thanks to the reviewers for the positive feedback and helpful comments for correction or modification.
We believe that the given suggestions have resulted in an improved revised manuscript underlined, which you will find uploaded alongside with this document. The manuscript has been revised to address the comments from reviewers, which are appended alongside our responses to this letter.
We very much hope that the revised manuscript will be accepted for publication in Microorganisms.
Sincerely yours,
Dongryeoul Bae
Mailing address: Research Institute of Pharmaceutical Science, College of Pharmacy, Gyeongsang National University, Jinju 52828, S. Korea.
Phone: +82-55-772-2416; E-mail: [email protected]
Min Gab Kim
Mailing address: Research Institute of Pharmaceutical Science, College of Pharmacy, Gyeongsang National University, Jinju 52828, S. Korea.
Phone: +82-55-772-2427; E-mail: [email protected]
Manuscript No: Microorganisms-1873710
We thank the reviewers for their comments and suggestions on our manuscript. The revised manuscript has been edited taking into the account of all suggestions by the reviewers. In addition, we changed table 1 to supplementary data table 1 and modified table 2 and 3. We focused on foodborne pathogens and their isolation and identification methods. Figure 2 for analysis of PFGE banding patterns of STEC isolates was added. E. coli O157:H7 was replaced to Shiga toxin-producing E. coli (STEC). We believe that the revised manuscript is suitable for publication in Microorganisms. Our responses to the reviewer’s questions are listed below:
Reviewer #2
Response: We appreciate your suggestions and comments. We revised and modified our manuscript as you indicated.
Comments to the Author
- Target genes for identifying pathogen type (table 2) need to be named. it is difficult to know what gene is being used based on the primer sequence alone.
We have added the gene names in the table as suggested. Ln 267. (We deleted table 1)
- line 24 and other locations in the manuscript - the authors refer to 283 pathogens were detected, though the wording here should be changed to clarify that 283 of the isolates were presumptive pathogens, not that 283 different pathogens were found.
We have changed as suggested. Ln 24-25.
- line 173-174 - please provide more details on why microbes such as Pantoea agglomerans are considered clinically important pathogens
We have deleted and rephrased as suggested. We focused on foodborne pathogens and clinically important pathogens associated with WHO global priority pathogens list of antibiotic-resistant bacteria. Ln 209-224.
- one of the stated goals of this work was to track pathogens from farm to distribution. the methods used do not allow for tracking of specific subtypes, which would be necessary to achieve this goal. On lines 237 to 240, the authors state that the pathogens with different ABX profiles are likely to be different subtypes, but is not clear why that is. if mobile elements are responsible for encoding the resistance, why would the assumption be that the subtypes would be different? the same MLST subtype could have or not have the same ABX profile.
We conducted systematic tracking of pathogens from farm to distribution via DNA fingerprinting of STEC strains using PFGE analysis. We rephrased as suggested. Ln 289-299.

Round 2
Reviewer 1 Report
The manuscript improved a lot by the PFGE analyses. Thank you for adding!